# Travel fosters tool use in wild chimpanzees

**Thibaud Gruber[1,2,3]\*, Klaus Zuberbühler[1,2,4], Christof Neumann[1]**

[1]Department of Comparative Cognition, University of Neuchâtel, Neuchâtel, Switzerland; [2]Budongo Conservation Field Station, Masindi, Uganda; [3]Swiss Center for Affective Sciences, University of Geneva, Geneva, Switzerland; [4]School of Psychology and Neuroscience, University of St Andrews, St Andrews, United Kingdom

**Abstract** Ecological variation influences the appearance and maintenance of tool use in animals, either due to necessity or opportunity, but little is known about the relative importance of these two factors. Here, we combined long-term behavioural data on feeding and travelling with six years of field experiments in a wild chimpanzee community. In the experiments, subjects engaged with natural logs, which contained energetically valuable honey that was only accessible through tool use. Engagement with the experiment was highest after periods of low fruit availability involving more travel between food patches, while instances of actual tool-using were significantly influenced by prior travel effort only. Additionally, combining data from the main chimpanzee study communities across Africa supported this result, insofar as groups with larger travel efforts had larger tool repertoires. Travel thus appears to foster tool use in wild chimpanzees and may also have been a driving force in early hominin technological evolution.

**\*For correspondence:** thibaud. gruber@gmail.com

**Competing interests:** The authors declare that no competing interests exist.

## Introduction

What evolutionary pressures have favoured tool use in some species, including chimpanzees and humans, but not others? Recent work in non-human primate species has focussed on the role of ecological variables for the emergence of tool use (*Fox et al., 1999*; *Humle and Matsuzawa, 2002*; *Möbius et al., 2008*; *Spagnoletti et al., 2012*; *Sanz and Morgan, 2013*). These studies have enlightened our understanding of how ecology influences animal culture (*Whiten et al., 1999*; *Laland and Janik, 2006*) and are also informative for modelling early hominin lifestyle (*Susman and Hart, 2015*).

For non-human primates, Fox and colleagues proposed three hypotheses to test the relation between ecological factors and the innovation of feeding-related tool use in primates, i.e., the 'invention', 'necessity', and 'opportunity' hypotheses (*Fox et al., 1999*). While the invention hypothesis states that new forms of tool use are rare chance events, which spread through social learning (*Fox et al., 2004*), the necessity and opportunity hypotheses state that ecological factors can have an additional influence (*Sanz and Morgan, 2013*). While the necessity hypothesis predicts that tool use emerges as a response to food scarcity, the opportunity hypothesis predicts this emergence as a response to encounters with either the material needed to make a tool or the resources to be extracted by the tools (*Koops et al., 2013*). The current literature has generated conflicting and inconclusive results concerning the different ecological hypotheses, even within the same species (chimpanzees (*Pan troglodytes verus*; necessity: *Yamakoshi, 1998*; opportunity: *Koops et al., 2013*; inconclusive: *Furuichi et al., 2015*); capuchins (*Sapajus* spp., necessity: *Moura and Lee, 2004*; opportunity: *Spagnoletti et al., 2012*); bonobos (*Pan paniscus*; inconclusive: *Furuichi et al., 2015*); see (*Sanz and Morgan, 2013*) for a review).

**eLife digest** There is currently much debate about the origins of animal culture, including why some animals have acquired the ability to use tools. Ecological problems often lead to the innovation of new tools. For example, a particular desirable food item may not be reachable without using a tool, or environmental conditions may make it difficult for an animal to find food without help.

Gruber et al. investigated how particular ecological factors influenced the use of tools in wild chimpanzees by combining controlled field experiments and observational data. When the ecological conditions were the most demanding, wild chimpanzees engaged most with the honey-trap experiment, an experiment where they had to use a tool to extract honey from a cavity dug in a log. Chimpanzees spent a longer time engaging with the apparatus when not much food was available and they had to travel more to obtain it. However, actual tool use during the experiments was only influenced by the travel effort made by the chimpanzees before they engaged with the log, not by how much fruit they had eaten beforehand.

In a larger analysis that included data from all of the long-term field sites with habituated chimpanzees, Gruber et al. found that chimpanzee communities that travel further on a daily basis use a wider range of tools to acquire food. These results suggest that travel is an important factor to consider when studying how tool use evolved. Furthermore, these results can be extrapolated to humans, who both travel further and use a greater variety of tools than chimpanzees.

Although innovation and culture are closely linked, innovation is mostly performed by individuals whereas culture is a social process. However, both are shaped by the environment. The next step will therefore be to disentangle and quantify the different contributions of environmental, individual and group factors in explaining how culture evolves.

Research on non-primates has generated an additional ecological hypothesis, the 'relative profitability hypothesis' to explain the emergence of tool use, which is based on optimal foraging theory and work on New Caledonian crows (*Corvus moneduloides*). This hypothesis states that tool use can develop as a strategy to obtain dietary components difficult to obtain without tools, but only if this is more profitable than non-tool-based strategies and as long as the ecological conditions, such as low predation pressure, allow it (*Rutz and St Clair, 2012*).

*Koops et al. (2014)* proposed an enlarged opportunity hypothesis, which includes not only ecological but also social and cognitive opportunities as drivers of tool use innovation and maintenance. In this view, necessity cannot explain tool use in animals because of the lack of correlations between selected environmental indicators and tool use. In particular, in a study with unhabituated chimpanzees of Nimba forest, Guinea, there was no correlation between fruit availability and ant remains in faeces, a proxy for stick use (*Koops et al., 2013*). Additionally, there was no relation between feeding-related tool use variants and the number of dry months across chimpanzee sites through Africa, further suggesting that tool use did not emerge out of necessity (*Koops et al., 2014*). In contrast, support for the necessity hypothesis comes from another study with the nearby habituated chimpanzees of Bossou, Guinea, where nut-cracking increased when fruit availability was low, suggesting that tool use is a fall back strategy during periods of food scarcity (*Yamakoshi, 1998*).

One explanation for these conflicting results is that the necessity hypothesis is difficult to test. For instance, *Sanz and Morgan (2013)* argue that the abundance of preferred food is a poor proxy for necessity and that even low levels of these foods may not be sufficient to trigger significant behavioural changes. Second, necessity may be driven by the lack of particular micronutrients essential for survival but that do not account for a major part of the diet (*Sanz and Morgan, 2013*). Necessity-based tool inventions, in other words, may not always function to compensate for low caloric intake. A third problem with the necessity hypothesis may also be due to the narrow focus of the analyses conducted to test it, e.g. feeding opportunities determined through phenological surveys, with no data on (a) whether animals actually seize these opportunities, (b) their variation across large time-scales (*Gruber et al., 2012a*), (c) the energetic costs incurred to benefit from them (*Pontzer and Wrangham, 2004*; *Lehmann et al., 2007*; *Amsler, 2010*), and (d) the differential needs of

individuals across time. In this respect, while analysing entire communities or populations can be useful, for instance by correlating phenological variables with tool use frequencies or tool repertoire sizes of entire communities (see below), individual needs may differ substantially within groups, suggesting that additional levels of analysis may be necessary to test the necessity hypothesis.

In this study, we were interested in the role of ecological factors in the emergence of chimpanzee tool use at the individual level. We studied how individuals of a chimpanzee community known for its limited tool use behaviour, the Sonso community of Budongo Forest (*Pan troglodytes schweinfurthii*), behaved in an experimental foraging task that required tool use. Although Sonso chimpanzees use tools in non-feeding contexts, such as for personal hygiene or communication, they have only been observed to use one type of tool to access resources, which consists of folding and chewing a handful of leaves to make a sponge, usually to collect water (*Reynolds, 2005*). Recently, some members of the Sonso community have learned a new technique, moss-sponging, to access mineral-rich suspensions from a clay pit (*Hobaiter et al., 2014*).

The Sonso community is part of a larger population of about 700 chimpanzees living in Budongo Forest, which most likely show the same limitations in tool use behaviour (*Gruber et al., 2012a*). For this reason, they constitute ideal subjects to study the emergence of new tool use behaviours, unlike other populations that already have complex food-related tool repertoires (*Whiten et al., 1999*).

We analysed data from a long-term field experiment, the honey-trap experiment, in which subjects were exposed to a novel foraging task that could only be solved with a tool. In doing so, we controlled for opportunity-based ecological aspects by presenting subjects with a standardised apparatus, which consisted of a small cavity drilled into a portable log, filled with liquid honey (*Gruber et al., 2009*, *2011*). Our goal was to test individuals under conditions of high ecological validity, over an extended period of time (2009–2015), with an unprecedented subject pool of over 50 individuals of a fully habituated community. In contrast to previous studies, our experimental approach allowed us to carry out analyses at the individual level, by comparing individuals in their interactions with the apparatus (*Gruber et al., 2009*, *2011*).

In our previous work, we found that 10 of 52 individuals (19.2%) who engaged with the apparatus proceeded to manufacture a leaf-sponge to extract artificially provided honey (*Gruber et al., 2009*, *2011*; *Gruber, 2016*). This behaviour is customarily used by wild chimpanzees to drink water, but there are no reports of chimpanzees using this behaviour to collect naturally available honey from bee nests. During our experiments, we also recorded two individuals using a stick to access the honey, but only after much exposure and experimental facilitation (see Material & Methods), and in contrast to another Ugandan chimpanzee community, where stick use was customary to access experimentally provided honey (*Gruber et al., 2009*, *2011*).

In the current study, we combined our long-term experimental data and behavioural observations to determine the natural parameters that influenced individual variation in engagement with the apparatus and the use of tools. As our experimental design controlled for opportunity, we were able to assess the influence of two key necessity-related variables, feeding time spent on ripe fruits (a proxy for food availability) and travel effort (a proxy for energetic demands), measured as the proportion of travel in the activity budget, on individuals' (a) engagement time with the apparatus and (b) probability of tool use. As we had no specific predictions concerning the relevant time intervals, we carried out these analyses incorporating data from different time periods prior to interaction with the apparatus.

Second, to determine whether any eventual patterns characterised chimpanzees as a species, we ran a cross-population comparison of travel behaviour and fruit feeding in relation to differences in food-related tool repertoires comparing data from all long-term chimpanzee communities. Finally, we discuss how our findings can shed light on the different hypotheses outlined above, and how they can contribute to a unifying model of the emergence of tool use.

## Results

We analysed a total of 292 experimental trials (N = 52 subjects, mean/median number of trials per individual: 5.6/3.0, range: 1–39). Mean engagement time with the apparatus was 111 s (N = 292 trials, range: 1–1275 s). In 21 of these trials (7.2%), subjects also used a tool. These cases were distributed over 16 different experimental days (11 with a single tool-user, five with two successive tool-users). For each trial, we determined the preceding travel and ripe fruit feeding behaviour of the

subject by systematically varying the time periods before each experiment (ranging from 1 to 13 weeks). To this end, we determined the proportion of all scans that contained travel and ripe fruit feeding for the focal individual of the test subject's party. This is a reasonable approach since members of a chimpanzee party typically engage in the same behaviour at a given time (see Material and methods).

The first model assessed how a subject's engagement time with the apparatus was related to ripe fruit feeding, travel time and time period. This model was significant overall (linear mixed-effects model, likelihood ratio test (LRT): $\chi^2 = 188.1$, df = 10, p<0.0001, $R^2_m = 0.33$, **Table 1**), with a significant three-way interaction between ripe fruit feeding, travel time and time period (LRT: $\chi^2 = 5.77$, df = 1, p = 0.0163, **Figure 1A**). Specifically, when subjects fed little on ripe fruits, they engaged more with the apparatus, provided they also travelled much. This effect was modulated by the duration the subject was recorded in the same condition. For example, chimpanzees engaged more with the apparatus if they had travelled more and had consumed less ripe fruits for longer than shorter periods of time (**Figure 1A**, lower panel). However, when subjects spent much time feeding on ripe fruits, there was less variation in time spent engaging with the apparatus, regardless of prior travel time. In addition, older individuals and males engaged less with the apparatus than young individuals and females.

Concerning the occurrence of tool use (observed in 21 of 292 trials; 7.2%), we built three generalized linear mixed models at three different time periods (1 week, 7 weeks and 13 weeks) because a single model analogous to the one presented above did not converge. Each of these models included the interaction between ripe fruit feeding and travel. We found that only the 1-week model was significantly different from its corresponding null model (LRT: 1 week: $\chi^2 = 12.0$, df = 5, p = 0.0346, $R^2_m = 0.30$; 7 weeks: $\chi^2 = 7.6$, df = 5, p = 0.1810, $R^2_m = 0.19$; 13 weeks: $\chi^2 = 8.5$, df = 5, p = 0.1299, $R^2_m = 0.18$). Contrary to the previous engagement time model, we did not find any effect of the interaction between ripe fruit feeding and travel time on the likelihood of tool use (all p>0.1, **Table 2**). However, we found a significant main effect of travel time on the probability of tool use, which increased with travel time (**Table 3**, **Figure 1B** top panel). No such result was found for ripe fruit feeding, although the effect went into the expected direction (i.e., more tool use with less ripe fruit feeding). For the other two time periods, the estimated effects of travel time and ripe-fruit feeding also went into the expected directions (**Table 3**, **Figure 1B**).

Finally, we analysed our data set on published estimates of diet and travel related behaviour of nine habituated wild chimpanzee communities (**Table 4**). In accordance with the results found in our analysis, we found that larger tool repertoires were associated with lower percentages of fruit consumption (Spearman's rho = −0.43, N = 9, **Figure 2A**) and higher percentages of travel (rho = 0.61,

**Table 1.** Results of LMM for the engagement of the Sonso chimpanzees with the honey-trap experiment. *p*-values for intercept and terms comprised in the three-way interaction are omitted. Reference levels for categorical predictors are female (sex), and no (tool use). *p*-values resulted from likelihood ratio tests.

| | $\beta$ | $\pm$ *se* | *t* | *p* | 95% CI |
|---|---|---|---|---|---|
| Intercept | 0.04 | 0.27 | 0.14 | | |
| Ripe fruit feeding | 0.04 | 0.05 | 0.80 | | |
| Time period | −0.00 | 0.01 | −0.11 | | |
| Travel time | 0.08 | 0.05 | 1.73 | | |
| Sex (male) | −0.31 | 0.40 | −0.78 | 0.4517 | −1.100, 0.473 |
| Age | 1.21 | 0.09 | 13.18 | 0.0000 | 1.028, 1.387 |
| Tool use (yes) | 1.25 | 0.12 | 10.66 | 0.0000 | 1.021, 1.481 |
| Auto correlation | −0.30 | 0.01 | −36.13 | 0.0000 | −0.313, −0.281 |
| Ripe fruit : Time period | −0.00 | 0.01 | −0.54 | | |
| Ripe fruit : Travel time | −0.02 | 0.02 | −1.32 | | |
| Time period : Travel time | 0.01 | 0.01 | 1.44 | | |
| Ripe fruit : Time period : Travel time | −0.02 | 0.01 | −2.40 | 0.0163 | −0.030, −0.003 |

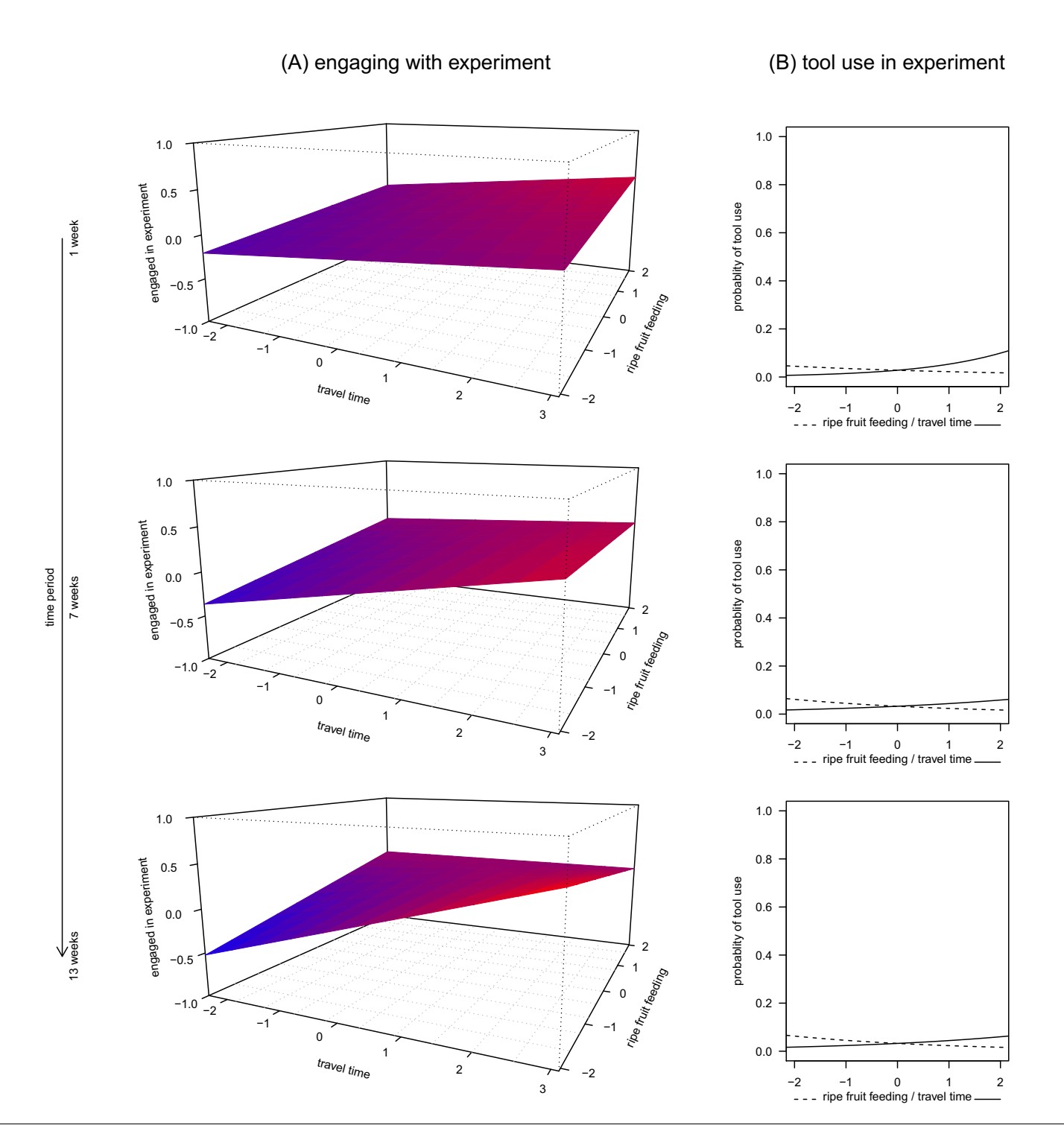

**Figure 1.** The relationship between ripe fruit feeding, travel time, time period and engagement in the honey experiment (A, *Figure 1—source data 1*) and ripe fruit feeding, travel time and tool use during the experiment (B, *Figure 1—source data 2*, *3* and *4*). Each panel shows the relationship between ripe fruit feeding, travel time and engagement, respectively use of tools, for time periods of 1, 7 and 13 weeks. All variables were standardized to a mean = 0 and SD = 1. For better readability, colour gradients along the model planes reflect predicted values along the vertical axis (engaged in experiment): larger values appear in red and smaller values in blue.

The following source data is available for figure 1:

*Figure 1 continued on next page*

*Figure 1 continued*

**Source data 1.** Engagement data.
**Source data 2.** Tool data 1 week.
**Source data 3.** Tool data 7 weeks.
**Source data 4.** Tool data 13 weeks.

N = 9, *Figure 2B*). When using average distance travelled per day, we found again a positive relationship with size of tool repertoire (rho = 0.77, N = 7, *Figure 2C*). Similar to our experimental data, the effect of the travel-related variables was larger than the effect of ripe fruit feeding.

## Discussion

Our results indicate that travel is directly related to the probability of tool use behaviour in wild chimpanzees. Our data first showed that the combination of low ripe-fruit availability and high travel effort increased their motivation to engage with a foraging problem that required tool use. Specifically, in situations of low fruit availability, the subjects spent more time engaging with the apparatus than at times of high ripe fruit availability, suggesting that they were possibly more inclined to explore alternative hard-to-get food possibilities. Our second finding was that tool use was mainly driven by short-term changes in daily travel, and less so by fruit availability. Specifically, tool use increased with increasing amounts of travel before the experiment, but this was mostly a short-term effect, up to one week prior to an experiment (*Figure 1B*, *Table 3*). Taken together, these results suggest that travel generates extra energetic costs in situations of low fruit abundance and that tool use is more likely to appear if ecological situations force chimpanzees to explore alternative feeding options in situations of high energy expenditures.

In this respect, tool use itself does not appear to be fostered by resource limitation, but rather by increased energetic costs. While tool use was interpreted as a fall back strategy in response to food scarcity in Bossou (*Yamakoshi, 1998*), in line with the original definition of necessity (*Fox et al., 1999*), this effect may not be observed in communities that do not display habitual feeding-related tool use behaviour, such as Sonso. The Budongo Forest has been described as a rich habitat where periods of extreme food scarcity are absent (*Newton-Fisher, 1999*), which may prevent chimpanzees from experiencing extreme necessity. Food availability nevertheless undergoes seasonal fluctuations (*Reynolds, 2005*) and, over the last decade, the food supply has noticeably gone down, in part due to anthropogenic activities (*Babweteera et al., 2012*). The Sonso chimpanzees have responded with behavioural adaptations to the disappearance of their original food resources (*Reynolds et al., 2015*), which suggests that detailed analyses are needed to better understand how food variation affects chimpanzee behaviour. Overall, our results suggest that chimpanzees are more eager to exploit difficult resources when the ecological conditions are more demanding relative to average conditions, both in terms of low food availability and high amounts of energy required to

**Table 2.** Likelihood ratio tests for full model and the interaction between ripe fruit feeding and proportion of travel time for the tool use models. Null models contained the random effects structure and the auto-correlation term.

| Time period | Full vs. null model (df = 5) | | Interaction Ripe fruit : Travel time (df = 1) | |
| --- | --- | --- | --- | --- |
| | $x^2$ | $p$ | $x^2$ | $p$ |
| 1 week | 11.99 | 0.0349 | 0.25 | 0.6169 |
| 7 weeks | 7.58 | 0.1810 | 0.02 | 0.8931 |
| 13 weeks | 8.52 | 0.1299 | 0.43 | 0.5116 |

**Table 3.** Model results for GLMMs testing the occurrence of tool use. *p*-values are presented only for the first model as the two other models were not significant (see **Table 2**). All numeric predictor variables were standardized to mean = 0 and SD = 1. For sex, 'female' is the reference level.

| | 1 week | | | | 7 weeks | | | | 13 weeks | | | |
|---|---|---|---|---|---|---|---|---|---|---|---|---|
| | $\beta$ | $\pm$ *se* | *z* | *p* | $\beta$ | $\pm$ *se* | *z* | *p* | $\beta$ | $\pm$ *se* | *z* | *p* |
| Intercept | −3.55 | 0.42 | −8.47 | 0.0000 | −3.39 | 0.37 | – | – | −3.40 | 0.37 | – | – |
| Ripe fruit feeding | −0.24 | 0.26 | −0.93 | 0.3525 | −0.33 | 0.24 | – | – | −0.35 | 0.24 | – | – |
| Travel time | 0.67 | 0.30 | 2.25 | 0.0242 | 0.30 | 0.27 | – | – | 0.32 | 0.26 | – | – |
| Sex (male) | −0.07 | 0.58 | −0.12 | 0.9062 | 0.04 | 0.54 | – | – | 0.04 | 0.55 | – | – |
| Age | −0.56 | 0.33 | −1.72 | 0.0855 | −0.56 | 0.29 | – | – | −0.55 | 0.29 | – | – |
| Auto-correlation | 0.80 | 0.18 | 4.44 | 0.0000 | 0.52 | 0.16 | – | – | 0.50 | 0.16 | – | – |

obtain the food. While high travel effort in itself is not necessarily linked with low diet quality in chimpanzees (e.g. *Riedel et al., 2011*), our analyses show that a combination of the two favours the exploration of alternative food resources, which creates opportunities for acquiring new tool behaviours. We interpret these findings as support for the more general idea that necessity can also drive invention in wild chimpanzees, when energetic demands are high. Necessity, in other words, is likely to be a major factor in driving the emergence of tool use behaviour in chimpanzees, if it is redefined to take into account both energetic costs and opportunities to compensate these costs. These results underline the importance of individual-based analyses that take into account data on both

**Table 4.** Data set for the cross-community comparison of nine wild chimpanzee study sites. Number of tools used were taken from *Sanz and Morgan (2007)*, except for Fongoli.

| Subspecies | Site/ group | Number of tools | % fruit in diet | % travel | Daily travel distance (km) | Reference |
|---|---|---|---|---|---|---|
| *verus* | Bossou | 13 | 60.3 | 19.5 | – | *Hockings et al. (2009, 2012)* |
| | Fongoli | 10* | 60.8 | 11.0 | 3.3* | *Bogart and Pruetz (2011); Pruetz and Bertolani (2009)* |
| | Tai/North | 11 | 85.0 | 22.0 | 3.7§ | *Boesch and Boesch-Achermann (2000); Boesch et al. (2006); Herbinger et al. (2001)* |
| *troglodytes* | Goualougo | 11 | 56.0 | 12.8† | – | *Morgan and Sanz (2006); Sanz† (2004)* |
| *schweinfurthii* | Gombe | 12 | 43.0 | 13.6 | 3.9§ | *Wrangham (1977)* |
| | Kanyawara | 2 | 66.6 | 11.0 | 2.1§ | *Pontzer and Wrangham (2004); Potts et al. (2011)* |
| | Mahale/M | 5 | 31.0 | 18.6‡ | 4.8¶ | *Huffman‡ (1990); Matsumoto-Oda¶ (2002); Nishida and Uehara, (1983)* |
| | Ngogo | 4 | 91.5 | 14.0 | 3.0# | *Amsler# (2010); Potts et al. (2011)* |
| | Sonso | 1 | 65.5 | 7.5 | 2.1** | *Bates and Byrne** (2009); Fawcett (2000); Newton-Fisher (1999)* |

\* Jill Pruetz, personal communication; travel estimate based on data from rainy season;

percentage of travel in daily budget:

† from her table 6.2, taking the highest value (range: 7.6–12.8) as travel activity was likely underestimated because of low habituation (*Sanz, 2004*, p.169);

‡ from his table 12.2, mean over individuals of both sex in the year 1985;

daily travel values:

§ average calculated across sex following *Pontzer and Wrangham (2004)*;

# from her table I, calculated as sum of hourly averages over a 10-hr activity day, based on males only;

¶ from her figure 4, calculated across seasons and sex;

** calculated from the average provided for each sex.

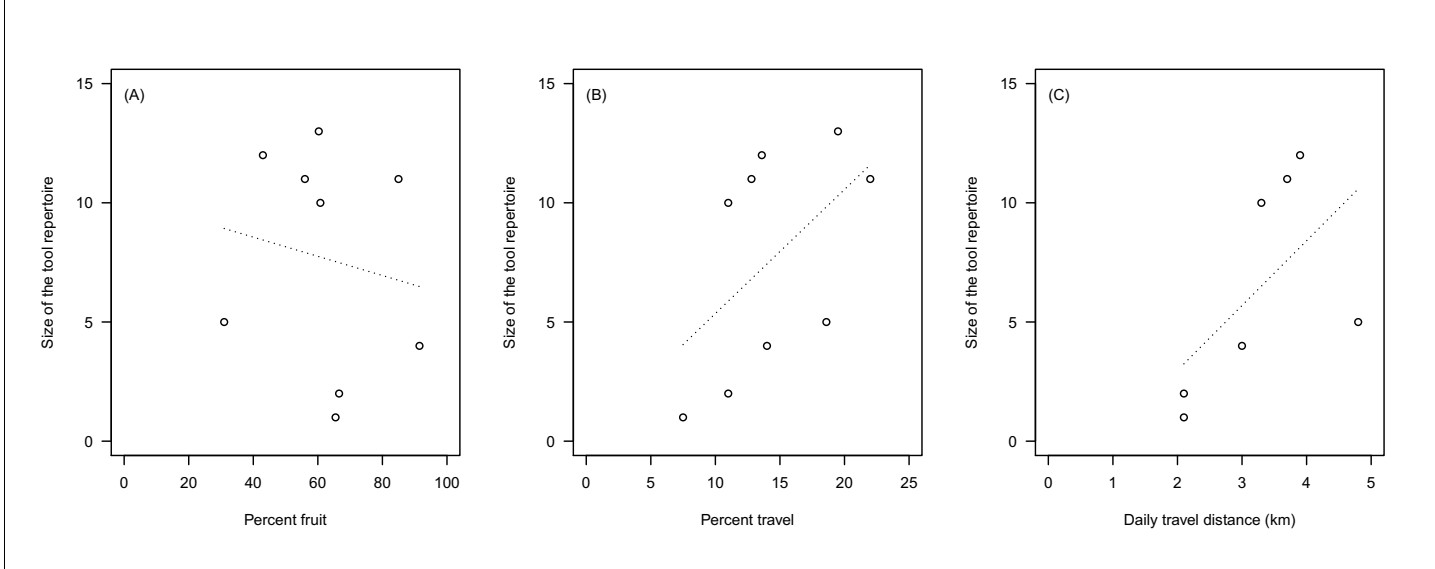

**Figure 2.** Relationship between percentage of fruit in the diet (**A**), percentage of travel in the activity budget (**B**), daily travel path (km, **C**), and the number of feeding-related tools described in currently documented long-term habituated chimpanzee communities. See *Table 4* for details.

energetic expenditure and intake, with potentially important implications for theories about the origin of tool use behaviour.

Our results are in line with the 'relative profitability hypothesis', which states that extractive tool use will occur if it is relatively more profitable than other alternative foraging strategies that do not rely on tool use (*Rutz and St Clair, 2012*). If increased travel effort represents an extra energetic cost, then tool use is a relatively more profitable strategy, especially if this occurs in ecologically challenging situations, which may trigger the switch from non-tool to tool-based foraging. Interestingly, the chimpanzees of Budongo Forest have increased their crop raiding habits over the last decade (*Tweheyo et al., 2005*), a probable response to a general decrease in food availability in the forest (*Babweteera et al., 2012*). As such, the innovation of novel tool use may only be one possible reaction to a changing environment, highlighting the flexibility of chimpanzees in dealing with changes in food availability (*Hockings et al., 2015*). Another facet of the relative profitability hypothesis is that tool use may provide individuals with a selective advantage over non-tool-using individuals (*Patterson and Mann, 2011*), as it provides them access to an energetically valuable resource, although in only 7% of trials did subjects succeed to do so. Perhaps this is not so surprising as tool use innovation is itself rarely observed in the animal kingdom (*Shumaker et al., 2011*) and only some species will develop tool use under identical ecological conditions (*Rutz and St Clair, 2012*), a reasoning that may apply at the population or individual level, as suggested by the current study.

While alternate strategies, such as crop-raiding, may contribute in part to the general lack of tool use inventions in this community, it is equally possible that psychological mechanisms can explain some of the observed patterns, offering insights into the 'invention hypothesis'. Here, one important result of our study is that the large majority of the tool-using individuals (19 of 21 cases, 90.5%) applied a familiar technique, leaf-sponging, in the experiment, behaving much different from when extracting honey from natural bee nests with their hands. Nevertheless, while adapting an existing behaviour to a novel context may be considered an innovation (*Reader and Laland, 2003*; *Reader et al., 2016*), only two individuals chose a different technique by attempting to use sticks. However, these two individuals did not incorporate this behaviour into their repertoire, raising questions about how wild chimpanzees represent artefacts as potentially useful tools (*Gruber et al., 2015*; *Gruber, 2016*). Additional studies are needed to explore the cognitive processes underlying chimpanzee tool use and, particularly, to decipher how ecological pressures and cognitive factors interact to lead to tool use innovation.

A neglected aspect in this study were the social opportunities for individuals to engage with the device or observe others to do so (see *Koops et al. (2014)*). In our study, engagement with the apparatus overlapped between social contexts (*Figure 3*), suggesting that the presence of others did not prevent subjects from engaging with it. However, it is less clear how the presence of others influenced the use of tools. Six individuals used a tool while being alone, seven others while in the company of family members, and eight in the company of other group members, to the effect that the current study cannot disentangle the relative role of social competition. Although tool-users spent more time with the apparatus and consumed more honey (*Gruber (2016)* and see *Table 1*), it is unlikely that this was because they monopolized the log. Rather, these individuals had developed a successful technique to recover the honey, compared to others who abandoned the apparatus earlier (*Gruber, 2016*). However, social influences are also in terms of social learning opportunities. As described elsewhere (*Gruber, 2016*), chimpanzees were generally tolerant to each other, but it is unclear whether they learnt from each other that leaf-sponge use was a suitable solution to extract honey. Social learning is a reasonable explanation for three individuals, but individual learning cannot be ruled out, largely because leaf-sponging was already part of their behavioural repertoire. Nevertheless, wild chimpanzees can learn socially from each other, even in a competitive context, and it is equally possible that this may even enhance social learning as it facilitates close observations of the novel behaviour (*Hobaiter et al., 2014*).

From our data, we conclude that the emergence of tool use in our group was due to a combination of necessity (energetic demands), opportunity (inaccessible honey) and relative profitability (lack of alternatives), suggesting that ecological and temporal aspects of resources availability as well as individual efforts all played a role (*Gruber, 2013*). While it is important to quantify the food available over the entire home range, it is also important to take into account the temporal variation of food availability and its consequences on the relative attractiveness of alternative foods simultaneously available to individuals, even for foods as attractive as honey. We concur with *Koops et al. (2014)* that individuals must be exposed to the right ecological opportunities, in our case the honey-trap apparatus, and that the probability of tool use may directly depend on the frequency with which they will encounter this challenge, a parameter we controlled for in our experiment. For tool acquisition and spread to appear, the right social settings may also have to be present (*Sanz and Morgan, 2013*), under the form of opportunities for close observation (*Hobaiter et al., 2014*). In our case,

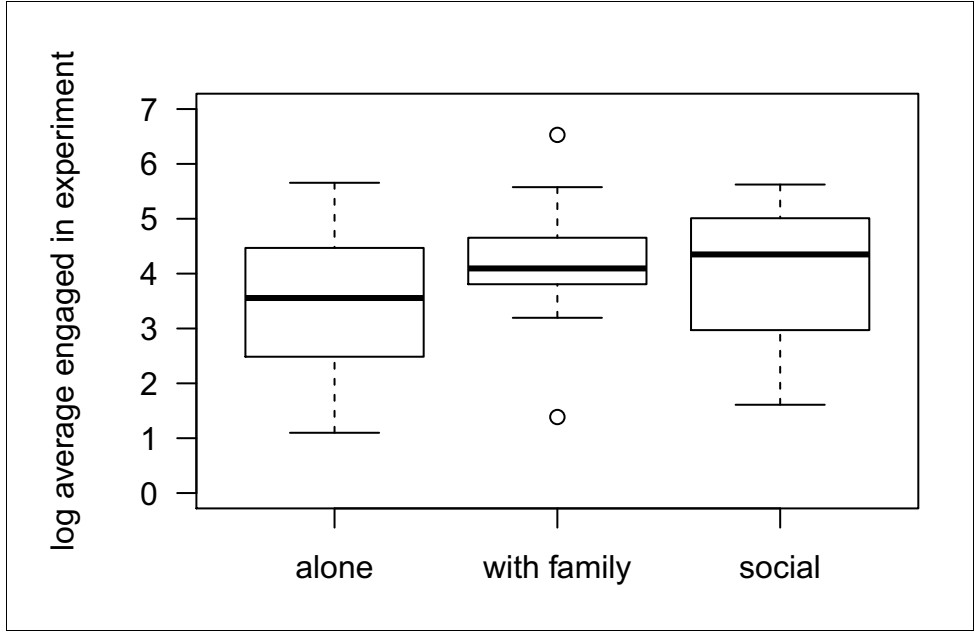

**Figure 3.** Range of engagement time of Sonso chimpanzees with the honey-trap experiment depending on the social context (alone, family-unit, or social).

encountering tools left by others (i.e. discarded sticks) did not appear to constitute a sufficient condition for social learning (*Gruber et al., 2011*). Our data also suggest that individual differences need to be taken into account. Finally, while cognitive abilities are likely to play a part in innovation and learning (*Gruber, 2016*), the emergence of tool use may also depend on whether it is relatively more profitable to do so, at any given time (*Rutz and St Clair, 2012*). Here, our data suggest that energetic demands resulting from individual variation in diet and travel effort directly influence the probability of tool use.

Is the relationship between travel and tool use generally found in the Hominidae? Our analysis of nine chimpanzee communities, although limited by the availability of published data on travel effort and tool use, suggests that our findings represent a general pattern. This analysis corroborated our empirical findings that travel effort and fruit consumption have opposing effects on tool repertoire size, and that travel effort, which arguably is best represented by the average daily distance walked by the chimpanzees, is likely to be more important than fruit consumption in explaining variation in tool repertoires between chimpanzee populations. In the long run, with more chimpanzee communities being currently habituated spanning across their entire ecological range (e.g. savanna in Fongoli, *Pruetz, 2006*), future studies will have to disentangle how environmental changes influence the relationship between tool use, energy intake and expenditure across a larger sample of chimpanzee populations. Regarding other great apes, both gorillas (genus *Gorilla*) and most orangutans (genus *Pongo*) show limited to no feeding-related tool use and interestingly they spend significantly less time travelling per day compared to chimpanzees, which suggests that their energy requirements are lower (*Pontzer and Wrangham, 2004*; *Pontzer et al., 2016*). Nevertheless, because travel is mostly arboreal in orangutans, more work is needed to estimate how this compares to chimpanzees, particularly with respect to Sumatran orangutans (*Pongo abelii*) for whom interesting variation in feeding-related tool use behaviour has been described (*van Schaik et al., 2003*; *Gruber et al., 2012b*).

The most promising comparison may come from the chimpanzees' closest relatives, the bonobos (*Pan paniscus*), where the lack of tool use has been connected to smaller travel distance between food patches and reduced feeding competition (*Wrangham and White, 1988*). The estimated daily travel effort for bonobos (2.6 km, *Furuichi et al. (2008)*) is comparable with some chimpanzee communities, incidentally the ones with the smallest tool repertoires for the species (Kanyawara, Ngogo and Sonso, all in Uganda), underlining a possible convergence in ecological pressures faced by these populations (*Gruber et al., 2010*). Interestingly, some convergence can also be found with modern humans. For instance, modern human hunter-gatherers walk on average 11.4–14.1 km/day (*Marlowe, 2005*; *Pontzer et al., 2012*, *2016*) and have the most diverse tool repertoire of all Hominidae, much beyond anything reported from the great apes (*Marlowe, 2010*). Combined, the results of the present study and the data from the three living hominines (*Homo, Pan*) reviewed here suggest an important role of travel in the emergence of tool use, but this needs to be tested across more study groups in different habitats and species. Whether this pattern holds for larger taxonomic groups beyond hominines remains to be investigated, taking into account the various ecological conditions faced by each species.

In conclusion, our findings suggest that tool use in hominids evolved in reaction to environmental changes that made preferred food harder to obtain. By extension, our results have direct implications for understanding hominid technological evolution, particularly in relation to the evolution of locomotor behaviour in the early stages of human evolution, as hominids faced similar ecological pressures. In effect, a number of major biogeographic events in the human lineage occurred at times of climate instability and it has been suggested that the development of tool use and sociality in hominins could constitute adaptive responses to heightened habitat instability (*Potts, 2013*). Australopithecines, for instance, evolved in a changing environment at the beginning of the Pliocene, where they faced more patchy resources of potentially lower quality (*Foley and Gamble, 2009*; *Potts, 2013*). Our findings support the view that tool use is connected to energy gain in a changing environment and that using tools is a response to increased costs of travel and lower quality of available food. In parallel, the adoption of bipedalism, which is less energetically costly than the quadrupedal and bipedal locomotion of chimpanzees, also allowed minimizing energy expenditure (*Pontzer et al., 2009*). Efficient, human-like bipedalism and tool use may have had complementary effects on travel costs, allowing both energy gain through the exploitation of novel ecological niches and energy economy during locomotion. Whether their development to unrivalled levels is what led

to the dispersal of early humans throughout Africa and the advent of complex technology around 3.0 million years ago (*Foley and Gamble, 2009*; *Harmand et al., 2015*) remains to be investigated.

## Material and methods

### Study site and subjects

The data were collected in the Sonso chimpanzee community of the Budongo Forest Reserve, Uganda (1°350–1°550 N, 31°180–31°420 E), at a mean altitude of 1050 m within 482 km$^2$ of continuous medium-altitude semi-deciduous forest (*Reynolds, 2005*). Rainfall in the Budongo Forest follows a bimodal pattern with two main rainy seasons between March and May and between September and November (*Figure 4A*, *Reynolds, 2005*). Habituation of the community started in 1990 with all residents identified, around 70 over the last eight years. The Sonso chimpanzees are notable for their complete lack of feeding-related tool-using behaviour with the exception of leaf- and moss-sponging (*Hobaiter et al., 2014*). Data included in the analysis consisted of six years of experimental data, collected between 2009 and 2015. We combined our experimental data with observational data collected between 2008–2015, up to three months before each experimental trial.

### The 'honey-trap' experiment

The Sonso chimpanzees are opportunists in relation to honey consumption, acquiring honey from natural bee nests (*Xylocopa* and *Apis* genus). Honey acquisition does not involve any tool use and is carried out with limited success only (T. Gruber, personal observation). In our honey-trap experiment, we provided subjects with the opportunity to systematically engage with a foraging problem, a 16 cm deep hole drilled into a 50 cm long log of about 25 cm diameter. The honey-trap experiment, by closely mimicking a natural setting, has proven its ecological validity, with over 80 individuals in two unhabituated and two habituated communities engaging with the experiment (*Gruber et al., 2009*, *2011*, *2012a*). The hole contained natural honey up to about 10 cm below the surface, which could only be extracted with the help of a suitable tool (*Gruber et al., 2009*, *2011*). Honeycombs were positioned so that they covered the hole to prevent insects, such as bees and ants, from entering it. Finally, a stick was potentially placed next to the log or directly plugged into the honey, depending on the experimental condition (*Gruber et al., 2011*). The apparatus was only set up when no chimpanzees were around and the experimenter (TG) always left the experimental area before the arrival of a subject. Several such apparatuses were in operation throughout the study period, all of them positioned at different locations throughout the Sonso territory. We never limited access to the apparatus, so that several individuals could participate during a given experimental day, possibly simultaneously.

Our final sample consisted of 292 trials, involving 52 individuals (over 70% of the total Sonso community), on 96 experimental days. In 124 cases of 292 (42.5%), the tested subjects were strictly alone while in 86 cases of 292 (29.5%) we tested individuals within a family unit. Finally, in 82 cases out of 292 (28%), other individuals joined the tested subject and also engaged with the honey-trap. These trials were also counted as engagement with the apparatus if the individual attempted to recover the honey (*Figure 3*). Experimental days were spread over six years (between 2009-2010 and 2012-2015, about 19 experimental days per year), with several weeks without experiment between each set of trials. Engagement time was defined as the time spent by a subject actively seeking to recover honey from the apparatus. Any attempt at playing with the log, or simply resting on the log was not included. In total, we observed 21 distinct tool use occurrences by 11 individuals: six in the alone context, seven in the family unit context and eight in the social context. For three of the latter trials, this occurred during social trials when other individuals had been using a tool before them. Because it has been shown that chimpanzee sponging is influenced socially (*Hobaiter et al., 2014*), we cannot exclude that these individuals may have been influenced by the previous individual engaging with the log. However, it is also possible that chimpanzees opted for a tool solution independently in each of these cases (see discussion in *Gruber (2016)*). For this reason, we considered each of the 21 instances independent from each other.

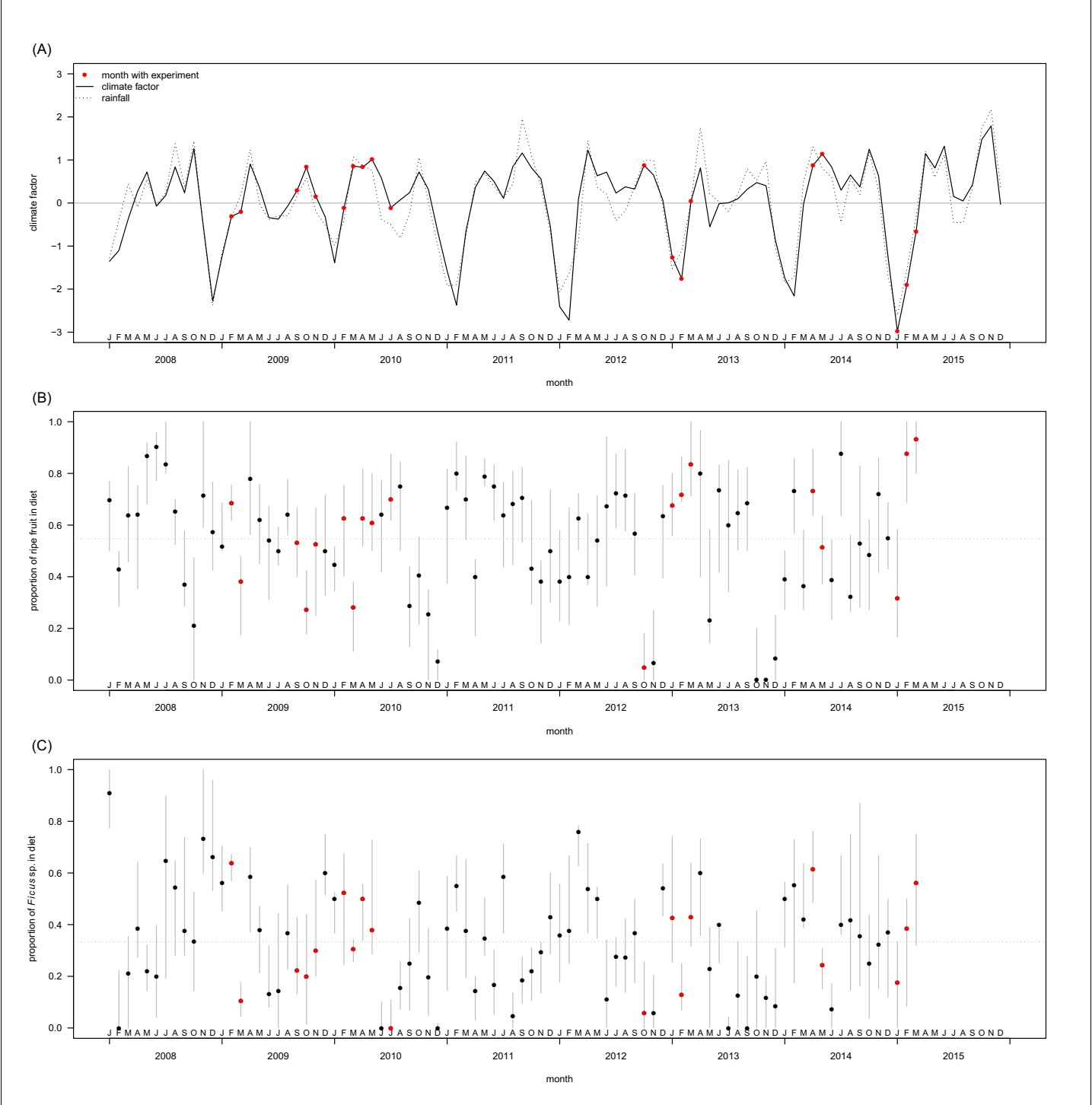

**Figure 4.** Temporal variation in climate in the Budongo Forest (**A**) and in feeding behaviour of the Sonso community (**B**, **C**) during the period covering the experimental trials. Months during which experiments were conducted are highlighted in red. (**A**) To define the climate factor, we calculated monthly cumulative rainfall and mean temperatures, extracted from daily values for rainfall, minimum temperature and maximum temperature in Budongo Forest available from 2001 through 2015 (Budongo Conservation Field Station long-term data 2001–2015). These monthly values were subjected to a principal component analysis (function 'princomp' in the stats package R v. 3.1.1, **R CoreTeam (2014)**). The climate factor corresponds to the scores of the first component of this analysis, which explained 64% of variance. Larger values along this axis correspond to larger values of rainfall, higher minimum temperature and lower maximum temperature as compared to smaller values along the climate factor. For reference, monthly cumulative rainfall is also plotted in this panel (dashed line). Both variables were standardized to mean = 0 and SD = 1. As such, values of 0 indicate average climate/rainfall (horizontal grey line). Out of 19 months with experimental days, 10 were characterised by above-average climate/rainfall and 9

*Figure 4 continued on next page*

*Figure 4 continued*
by below-average climate/rainfall. (**B**) Variation in ripe fruit feeding behaviour. Shown are monthly median values of the proportion of ripe fruit in the diet for individuals that were observed at least five times feeding during a given month. Grey bars indicate quartiles and the horizontal dashed line represents the mean value across all individual-months. (**C**) Variation in fig feeding. Shown are monthly median values of the proportion of figs in the diet for individuals that were observed at least five times feeding on ripe fruit during a given month. Grey bars indicate quartiles and the horizontal dashed line represents the mean value across all individual-months.

## Influence of seasonality and identification of periods of food scarcity in Budongo Forest

Experiments were carried out both in dry and wet seasons, to control for a potential effect of seasonality and to encompass the entire range of ecological variation in terms of possible food offer available to the chimpanzees (*Figures 4* and *5*). Over the last decades, Budongo Forest has been

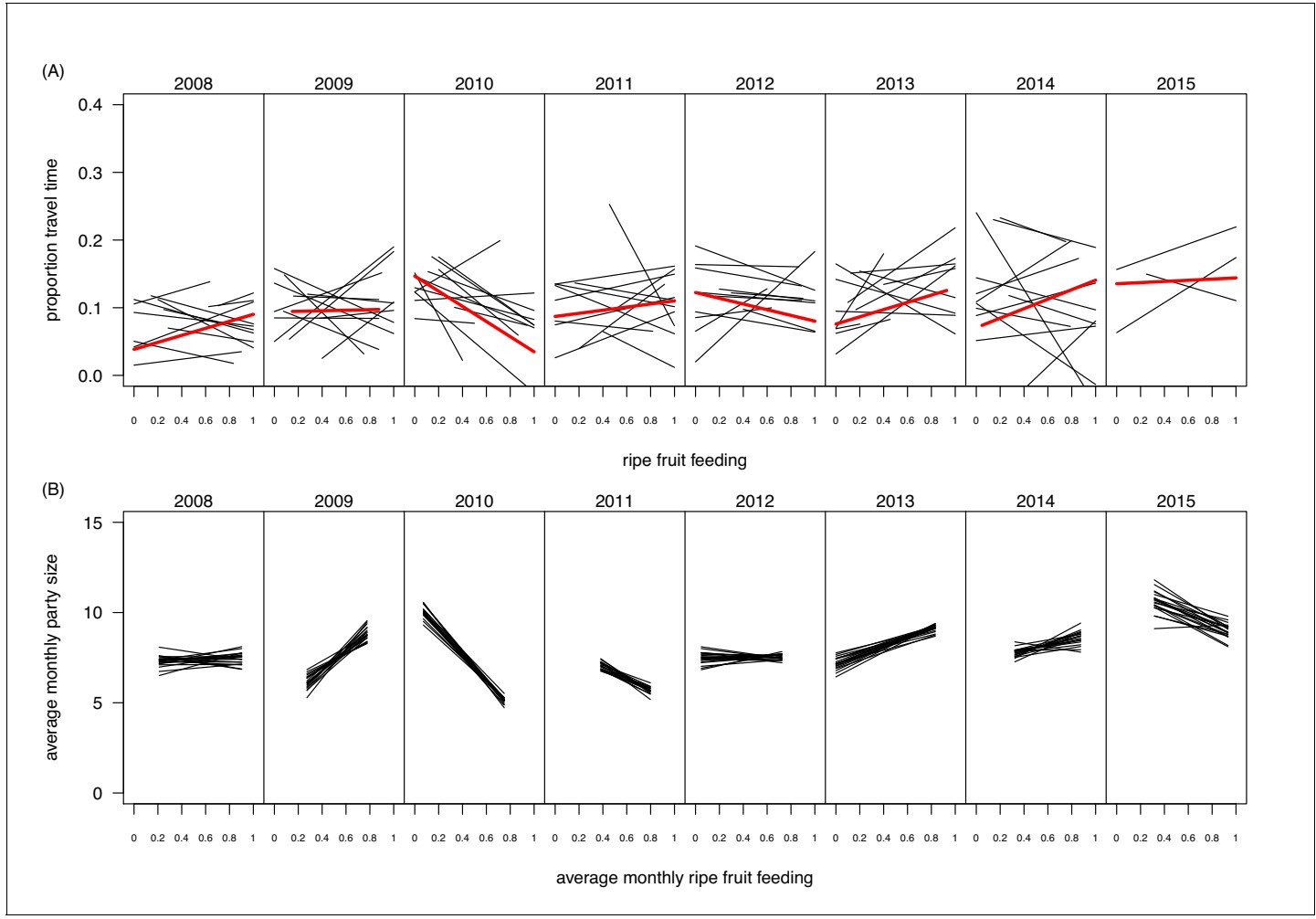

**Figure 5.** Within- and between-year variation in the relationships between ripe fruit feeding and (**A**) travel time and (**B**) mean monthly party size. In (**A**), each black line represents the regression line of travel time on ripe fruit feeding within a month, based on data from focal individuals. Thus, per panel 12 lines are depicted, except for 2015 for which data were available only for the first three months. The red line depicts the average regression over the respective year. In (**B**), each line represents a regression line of monthly average party size on average monthly ripe fruit feeding proportion. Each line is based on data from a random selection of parties (limited to one party per day) to calculate the monthly average party size. The randomization was repeated 20 times, resulting in 20 regression lines per panel. The panel for 2015 is based on regressions with only three data points as data were only available for the first three months of 2015.

described as a rich environment where chimpanzees do not face periods of food scarcity comparable to the ones experienced by other chimpanzee populations (*Newton-Fisher, 1999*). For instance, a study conducted during this period found that there was no positive relationship between food availability and party size, a marker of food scarcity (*Newton-Fisher et al., 2000*). Nevertheless, recent research has shown that the food supply has steadily decreased in Budongo Forest, suggesting the possible appearance of periods of food scarcity for the resident primate populations (*Babweteera et al., 2012*). Interestingly, when correlating party size with proportion of ripe fruit feeding over the whole duration of our study, we found an overall relationship close to 0, reflecting the results of the earlier study by *Newton-Fisher et al., (2000)*. However, we saw a large variation between years, with years (2009, 2013, 2014) where the relationship follows the more conventional chimpanzee pattern (i.e. more ripe fruit feeding coincides with larger parties), years (2010, 2011) where this pattern follows an opposite direction, and years (2008, 2012) where there is no clear pattern (*Figure 5B*). Additionally, even within a particular year, we observed substantial variation in monthly ripe fruit consumption (range: 0.00 to 0.93, *Figure 4B*). Similarly, there was also large variation in time spent feeding on fig species across the duration of our study (*Figure 4C*), with fig species often considered a fallback food for chimpanzees, and their consumption a potential marker of food scarcity (*Marshall and Wrangham, 2007*; *Harrison and Marshall, 2011*). Our experiments, spread across this spectrum, thus allowed us to test the potential effect of food scarcity and travel effort across a large range of ecological situations.

## Observational data

Long-term data on party composition as well as foraging and ranging behaviour have been collected by trained field assistants since the beginnings of the project. During focal animal follows, the field assistants note every 30 minutes a focal individual's activity (feeding, travel, resting, grooming) and, if feeding, the plant species and the plant part (ripe fruit, unripe fruit, leaves, flowers, bark) consumed. In addition, party composition is recorded by noting all adolescent and adult individuals in the focal animal's party. Data for dependent juvenile individuals are extracted from their mother's behaviour. To increase our sample on feeding and travel behaviour of individuals, we assumed that all party members expressed the same behaviour as a party's focal individual. This approach is justified given an analysis of a subset of our data for which the activity for all party members (in addition to the focal individual) was recorded. Across 31,278 party scans, the mean proportion of individuals that expressed the same behaviour as the party's focal individual was 0.8 (median = 1.0, range: 0.0–1.0).

For each subject who participated in an experimental trial, we calculated separately its time spent feeding on ripe fruit and its time travelling in the following way. We identified all data points in our behavioural database in which the subject was present in an observed party (regardless of whether the subject was the focal animal or not, see above). We then noted the respective focal animal's activity and plant part eaten. In other words, we considered the focal animal as representative for the experimental subject as long as they were part of the same party. Because juveniles who engaged with the experiments were still dependent to their mother at the time of the experiment (and therefore are not considered as individual points in the database), we extracted these data from their mother's data. For this study, we analysed N = 40,908 data points collected by nine experienced field assistants between 2008 and 2015.

From this database, we calculated ripe fruit feeding and travel time as proportions, i.e. as the number of data points feeding on ripe fruit relative to all data points spent feeding, and travelling relative to all observations of that subject (or its respective focal animal, if the subject was in the party but not itself the focal animal, see above). Because we had no a priori expectation as to what time period was meaningful to the chimpanzees, we considered different time periods, ranging from one week prior to the experiment up to 13 weeks before the experiment (i.e. approximately three months), using one-week increments. Note that we did this in a cumulative fashion, i.e. a given 2-week data point included the data of the first week before an experiment, a given 3-week data point included data from weeks 1 and 2, and so forth. We controlled for this inter-dependence statistically (see below). In this way, we assembled a total of 292×13 = 3,796 data points. Out of these, we had to exclude 52 data points because no observational data were available for a given subject (mostly during the shorter time periods). Our final data set comprised 3,744 data points, including data from 50 subjects that participated in the honey experiment.

## Data analyses

Our objective was to investigate how ecological parameters (feeding on ripe fruits and travelling) influenced the time the subjects engaged with the apparatus and whether a tool was used during the experiment. Our main predictor variables were the proportions of time spent feeding on ripe fruit and time spent travelling, plus their interaction. Further, we included a 3-way interaction between ripe fruit feeding, travel time and time period, reasoning that any effect of feeding and/or travelling may be short or long term. The time period variable indicated the number of weeks (range: 1–13, i.e. about three months) over which an individual's travel and feeding data were accumulated prior to an experimental trial.

### Engagement model

For engagement time, we used a linear mixed model with Gaussian error distribution and identity link (*Baayen, 2008*; *Bolker et al., 2009*). Apart from the 3-way interaction, which also comprised the two-way interactions and corresponding main effects, we included a subject's age (calculated from birth dates) and sex and whether or not a tool was used as additional fixed effects in the model. Subject ID and experimental trial ID were fitted as random intercepts. Following *Barr et al. (2013)*, we included random slopes, specifically ripe fruit feeding and travel proportions within subject ID and tool use (yes/no) within experiment ID. Prior to model fitting, all numeric predictors and engagement time were transformed (square root or log) and subsequently standardized to mean = 0 and SD = 1 (*Schielzeth, 2010*).

After fitting the initial model, we calculated an auto-correlation term to account for the temporal dependence of data points brought about by our measuring ripe fruit feeding and travel time at different time periods. To deal with this potential problem, we followed procedures developed by Mundry and collaborators (e.g. *Fürtbauer et al., 2011*; *Hedwig et al., 2015*). Starting with the residuals from the full model, for each data point we calculated the average of the residuals of all other data points of the same individual. These residuals were weighted by their time lag (i.e. weeks) with respect to the original data point. Following *Fürtbauer et al., (2011)*, the weight was normally distributed with a standard deviation determined by minimizing Akaike's information criterion of the full model that included the term as additional fixed predictor variable.

We ran model diagnostics following *Quinn and Keough (2002)*. We checked residuals for normality and homogeneity inspecting the histogram of residuals and a plot of fitted values versus residuals. We calculated variance inflation factors from a linear model excluding the random effects structure using the vif function from the car package (*Fox and Weisberg, 2011*). All variance inflation factors were smaller than 1.16, which was deemed unproblematic (*Field et al., 2012*). After including the auto-correlation term into our full model, we tested this full model against a null model (*Forstmeier and Schielzeth, 2011*), which comprised only the auto-correlation term and the random effect structure using a likelihood ratio test (*Dobson, 2002*; *Quinn and Keough, 2002*).

To assess statistical significance of single terms, we used likelihood ratio tests that compared nested models. For example, to test the 3-way interaction, we compared the full model (which included this 3-way interaction) against a model from which the 3-way interaction was removed but which still contained all lower-level terms comprising any of the three variables, i.e. the three 2-way interactions (ripe fruit feeding : travel time, ripe fruit feeding : time period, travel time : time period) and the three main effects (ripe fruit feeding, travel time, time period).

To assess model stability, we refitted the full model repeatedly, each time excluding one individual from the data set. There were no influential individuals with respect to the significance of the full model, i.e. our model was stable with respect to our entire set of predictor variables significantly explaining how much time individuals spent engaging with the honey experiment. However, this analysis also indicated one individual having been disproportionately influential, such that with this subject excluded from the data set the 3-way interaction was not statistically significant anymore. After removing the 3-way interaction (ripe fruit feeding : travel time : time period) and the two 2-way interactions including time period (i.e. time period : ripe fruit feeding and time period : travel time, assessed with likelihood ratio tests resulting in p>0.05), only the interaction between ripe fruit feeding and travel time remained significant. Note that the model with this individual excluded was still significantly different from its respective null model. The resulting effect of the ripe fruit feeding : travel time interaction resembles mostly what we found at short time periods of our full model (see

*Figure 1*). This suggests that we may have overestimated the effect of time period in our main analysis, but also suggests that the interaction between ripe fruit feeding and travel time is robust as far as influential individuals are concerned, which corroborates our main finding, i.e. that the time subjects engaged with the honey experiment was explained by the interaction of ripe fruit feeding and travel time.

## Tool use model

To test whether a tool was used or not during a trial, we used a generalized linear mixed model with binomial error and logit link function (*Baayen, 2008*; *Bolker et al., 2009*). Initially, we attempted to fit an equivalent model as for the engagement time analysis. However, this model did not converge, presumably because tool use was generally rare and our model was therefore too complex. Instead, we fitted three separate models at time periods of 1, 7, and 13 weeks, i.e., we excluded time period as predictor variable. In addition to subject age and sex, we included the 2-way interaction between ripe fruit feeding and travel time (as in the engagement model), and fitted engagement time as an offset term (*Fox and Weisberg, 2011*). Subject ID and experiment ID were added as random intercepts. All numeric predictor variables were transformed (square root or log) and subsequently standardized to mean = 0 and SD = 1. As for the engagement model, we first assessed the significance of the three full models (comprising all predictor terms, including the two-way interaction) versus the corresponding three null models (only comprising the random effects and the auto-correlation term) with likelihood ratio tests. Only if such a comparison revealed statistical significance did we explore the full model. To do so, we assessed the significance of the interaction term, which we removed if not significant (at alpha = 0.05) to allow interpretation of main effects (*Hector et al., 2010*; *Mundry, 2011*).

The largest variance inflation factor in any of the three models was 1.22, suggesting collinearity not to be problematic (*Quinn and Keough, 2002*). We checked for influential individuals in the same manner as described above, though only for the 1-week model. With regards to the significance of the full model, we found that in seven cases (i.e. seven different individuals), the likelihood ratio test for the comparison of the full against the null model revealed *p*-values larger than 0.1. Out of these seven overly influential individuals, six were individuals that were observed as having used tools at least once. Given that the overall number of tool uses was small (21 cases) compared to the total number of cases (N = 292) it is not surprising that excluding individuals that contributed to the number of tool uses pulls the model substantially towards the null model, i.e. tool use was random. However, in all models the parameter estimates for travel time were positive (mean = 0.38, range: 0.25 − 0.61), while all estimates for ripe fruit feeding were negative (mean = −0.29, range: −0.38 – −0.13), which is consistent with our finding that tool use was driven in separate directions by ripe fruit feeding and travel time, though the actual magnitude of these effects remains to be further investigated.

The engagement and tool use models were fitted with the lmer and glmer functions of the lme4 package (v. 1.1–7, *Bates et al. [2014]*) in R (v. 3.1.1, RRID:SCR_001905, *R Core Team [2014]*). We calculated marginal $R^2$ following *Nakagawa and Schielzeth (2013)* and *Johnson (2014)*.

## Cross-community comparison

Finally, we searched the published literature for estimates of the travel and feeding behaviour of wild chimpanzees. In particular, we collected data on activity budget, tool use and diet from long-term habituated chimpanzee communities for which tool-use behaviour was known (N = 9, *Table 4*). When possible, we used fruit consumption and travel data from the same study, as this would directly connect the travel effort with the food consumed at the time of the study. When this was not possible, we extracted or calculated the values from the literature. If there were more than one value for any of the variables, we selected the values that had been estimated the closest to each other. For tool use, we only took into account feeding-related tool use behaviour, as reviewed by *Sanz and Morgan (2007)*. We used estimates of travel in activity budget and proportion of fruits in the diet to compare with our experimental data. We calculated non-parametric (Spearman) correlations between these values and the number of different tools used in the respective communities. We also ran an additional correlation between number of tools and daily travelled distance when these data were available.

## Acknowledgements

The research leading to these results has received funding from the People Programme (Marie Curie Actions) and from the European Research Council under the European Union's Seventh Framework Programme for research, technological development and demonstration under REA grant agreement N°329197 awarded to TG, ERC grant agreement N°283871 awarded to KZ. We thank Uganda Wildlife Authority, Uganda National Council for Science and Technology and National Forestry Authority for allowing us to work in the Budongo Forest. We thank the Royal Zoological Society of Scotland (RZSS) for providing the core funding to the Budongo Conservation Field Station (BCFS) and all BCFS field assistants as well as the maintainers and contributors to the BCFS long-term database, in particular Coco Ackermann and Geresomu Muhumuza. We thank the reviewing editor, Herman Pontzer and an anonymous reviewer for their useful comments on the manuscript.

## Additional information

### Funding

| Funder | Grant reference number | Author |
| --- | --- | --- |
| European Commission | 329197 | Thibaud Gruber |
| European Commission | 283871 | Klaus Zuberbühler |

The funders had no role in study design, data collection and interpretation, or the decision to submit the work for publication.

### Author contributions

TG, Conception and design, Acquisition of data, Analysis and interpretation of data, Drafting or revising the article; KZ, Drafting or revising the article, Contributed unpublished essential data or reagents; CN, Analysis and interpretation of data, Drafting or revising the article

### Author ORCIDs

Thibaud Gruber, http://orcid.org/0000-0002-6766-3947
Christof Neumann, http://orcid.org/0000-0002-0236-1219

### Ethics

Animal experimentation: Permission to conduct the chimpanzee research was given by Uganda Wildlife Authority (UWA, permit FOD/33/02 to TG) and Uganda National Council for Science and Technology (UNCST, permit ns431 to TG). Research protocols were reviewed and approved by the veterinary staff at Budongo Conservation Field Station. Ethical approval was given by the Ethics Committees at the School of Psychology, University of St Andrews and the University of Neuchâtel.

## Additional files

### Supplementary files

• Source code 1. Datasets and model specifications.

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
