## [Decision Letter]

Thank you for submitting your article "The role of travel in chimpanzee tool use and hominidae technological evolution" for consideration by *eLife*. Your article has been reviewed by two peer reviewers, and the evaluation has been overseen by a Reviewing Editor and Eve Marder as the Senior Editor. The following individual involved in review of your submission has agreed to reveal their identity: Herman Pontzer (Reviewer #2).

The reviewers have discussed the reviews with one another and the Reviewing Editor has drafted this decision to help you prepare a revised submission.

The reviewers congratulate the authors on an interesting experiment that brings together an unusually large data set to investigate the drivers of food-related tool-use in chimpanzees. The manuscript is clearly written and the experiment judged a useful contribution to the field. The reviews address a broad range of concerns both in methods and interpretation that are summarized here, but agree that a revision that addresses their concerns would be worthy of further consideration. Finally, the title would be more accurate as: "The role of travel in chimpanzee tool use".

1) Perhaps most importantly, the data are analyzed to test only the 'necessity' hypothesis, which proposes that the emergence of tool-use is related to food scarcity and the need to access new food resources. However, (at least) three main hypotheses exist to explain the emergence of food-related tool use in primates. Namely, invention, necessity and opportunity hypotheses have all been proposed as possible explanations. The existing dataset could shed light on all of these hypotheses, strengthening the contribution of this work to our understanding of this interesting and important phenomenon. This is especially true since, as currently set out the paper's Introduction, the 'necessity' hypothesis does not appear to be a leading hypothesis to explain the emergence of tool-use in primates.

2) The authors claim to relate individual patterns of tool-use/engagement with their experimental apparatus to individual-level measures of fruit consumption and travel time. However, in the last paragraph of the subsection “Observational data” of the Materials and methods section, they clarify that they don't actually have individual level data but rather use data from a focal individual in the same party as the individual of interest and assume that their behaviors were the same. No quantitative demonstration that this is a reasonable assumption is presented, nor any justification given. It appears that the data are available to the authors to test their hypothesis, it is certainly not 'individual-level' data in the way most people conceive of the term, and this needs to be justified and clarified in the main text of the manuscript, rather than described in the Materials and methods section at the end.

3) The authors' need to clarify if experimental subjects were tested alone (as suggested in the fifth paragraph of the Introduction), and if so, how this was achieved given the description provided in the last paragraph of the subsection “The ‘honey-trap’ experiment”, which seems to indicate that other group members may have been present.

4) The Sonso chimpanzee community in Budongo have an unusual ecology-showing little evidence of a true low-food availability season (Newton-Fisher 1999). Some key behavioral responses of chimpanzees to low food availability also appear to be absent at Budongo, including the positive relationship between food availability and party size (Newton-Fisher et al. 2000). At many other study sites, chimpanzees respond to decreases in food availability via increased sub-grouping. This raises the question of whether the authors have actually tested patterns of tool-use under true periods of energetic stress. The fact that% time traveling and% fruit in diet were not significantly collinear in this study (subsection “Tool use model”, second paragraph), suggests that they may not have, as chimpanzees tend to range further when fruit availability is low. Given these peculiarities of their study community, the authors need to address more clearly why they believe they tested the 'necessity hypothesis' (i.e. sampled true periods of resource scarcity), as opposed to, sampling one part of what is actually an inverted U-shaped distribution.

5) How do the authors know that longer travel times are in fact evidence of energy stress? While travel will have some inherent energy cost, chimpanzees may be more active when food is more plentiful, and may be more likely to pursue attractive but less energy-rich foods when food energy is plentiful. For example, it has been suggested that chimps hunt more often when food availability and group sizes are high. Recent work from Tai forest (Riedel 2011) finds that high ranking females are more gregarious, eat a higher quality diet, and travel further than low ranking females. These studies suggest that high investment in travel is not necessarily a marker of energetic need. Similarly, because of the ecology of the Sonso community, it is not necessarily clear that the% of time spent feeding on ripe fruit is necessarily indicative of energetic need. In fact, if fruit is abundant, it may take individuals less time to meet their energetic needs. Demonstrating an increase in the amount of time spent feeding on fallback foods would provide more convincing evidence of energetic stress. Basically, you need to convince the reader that your proxies for energetic need are valid.

---

## [Author Response]

The reviewers congratulate the authors on an interesting experiment that brings together an unusually large data set to investigate the drivers of food-related tool-use in chimpanzees. The manuscript is clearly written and the experiment judged a useful contribution to the field. The reviews address a broad range of concerns both in methods and interpretation that are summarized here, but agree that a revision that addresses their concerns would be worthy of further consideration. Finally, the title would be more accurate as: "The role of travel in chimpanzee tool use".

We have modified the title, taking out the evolutionary claim as suggested by the reviewers, but would like to propose “Travel fosters tool use in wild chimpanzees” as a substitute title.

1) Perhaps most importantly, the data are analyzed to test only the 'necessity' hypothesis, which proposes that the emergence of tool-use is related to food scarcity and the need to access new food resources. However, (at least) three main hypotheses exist to explain the emergence of food-related tool use in primates. Namely, invention, necessity and opportunity hypotheses have all been proposed as possible explanations. The existing dataset could shed light on all of these hypotheses, strengthening the contribution of this work to our understanding of this interesting and important phenomenon. This is especially true since, as currently set out the paper's Introduction, the 'necessity' hypothesis does not appear to be a leading hypothesis to explain the emergence of tool-use in primates.

We thank the reviewers for this comment and apologize for letting them assume that the data we present in this article are only related to the necessity hypothesis. We agree with the reviewers that these data can potentially shed light on all of the hypotheses proposed to explain the emergence of tool use. In this revised version, we have therefore assessed our results through the different perspectives offered by these alternative but not necessarily mutually exclusive hypotheses. We have therefore established this goal in the Introduction (last paragraph).

In the Discussion, we analyse successively how our data can contribute to all the proposed hypotheses, before highlighting that they are best explained through a combination of these hypotheses. First, we have expanded our discussion of the relative profitability hypothesis (Discussion, third paragraph). We propose that if travel effort can indeed be considered an extra energetic cost, it may constitute an appropriate trigger to the switch from a non-tool-using strategy to a tool-using strategy. This switch occurs at a time when the primary non-tool-using strategy (non-tool-using fruit foraging) is not providing an optimal response to the energetic needs of individuals. Additionally, the relative profitability hypothesis is also supported by the fact that some individuals (the tool-users) will gain an advantage compared to the non-tool-users, because they can exploit a resource that others cannot (in the aforementioned paragraph).

This argument triggers our discussion of the ‘innovation hypothesis’. First, we have expanded our discussion of whether leaf-sponging used in the context of the honey-trap experiment can be considered novel (Discussion, fourth paragraph). We discuss the concept of innovation associated to these data, and conclude that the use of an old technique in a novel context can effectively be considered an innovation, particularly if it was never seen in the community before (see Reader et al. 2016). Additionally, we discuss shortly the cognitive demands associated with the innovation hypothesis as originally defined by Fox et al. (1999), although we do not expand too much on this question as it was already the focus of other publications (in the aforementioned paragraph).

Our discussion of the invention hypothesis also connects with the social opportunities that are given to chimpanzees to innovate or to learn novel behaviour (Discussion, fifth paragraph). In our study, we address this point by observing that the range of engagement time with the log largely overlapped across social settings (alone, family unit or social; displayed in the new Figure 3), suggesting that any competition between individuals to engage with the log did not hinder their opportunities to interact with it. Similarly, the possible innovation or learning of tool use during the experiment cannot be decisively attributed to the social opportunities that allowed an individual to access the apparatus, or to observe others engaging with the apparatus using a tool (in the aforementioned paragraph). Here, a different study focused on the spread of a novel behaviour, moss-sponging, in the Sonso community (Hobaiter et al. 2014) allow us to expand over social opportunities to learn novel tool use behaviour (in the aforementioned paragraph).

Finally, in an attempt to summarize the different aspects covered in each of the preceding sections, we propose that a combination of the four hypotheses explain best our data (Discussion, sixth paragraph). In essence, while opportunities to face a particular ecological problem are mandatory to trigger the use of tools, they have to be considered through a larger scope taking into account the current needs of an individual, also integral to the probability of developing tool use. As such, one must take into account whether it is relatively more profitable for the animal to exploit this resource with a tool, in comparison with non-tool-using options. The non-tool-using options, possibly less costly, can prevail for both potential innovators and learners if there are no major incentives to develop a novel behaviour (in the aforementioned paragraph).

2) The authors claim to relate individual patterns of tool-use/engagement with their experimental apparatus to individual-level measures of fruit consumption and travel time. However, in the last paragraph of the subsection “Observational data” of the Materials and methods section, they clarify that they don't actually have individual level data but rather use data from a focal individual in the same party as the individual of interest and assume that their behaviors were the same. No quantitative demonstration that this is a reasonable assumption is presented, nor any justification given. It appears that the data are available to the authors to test their hypothesis, it is certainly not 'individual-level' data in the way most people conceive of the term, and this needs to be justified and clarified in the main text of the manuscript, rather than described in the Materials and methods section at the end.

We agree that our introduction of individual patterns may have been confusing. Because our goal was to analyse the test subject’s behaviour over a long period (up to 13 weeks) before an experimental trial, we sought the best way to accurately represent its behaviour across this time. While using the individual focal data would have been the most accurate, it was also limiting the scope of our analysis. As there were around 70 individuals in the Sonso community at the time of the experiments, the likelihood that a test subject would be followed in the days preceding an experimental trial depended on the duration considered: the longer the time period considered was, the more chances there were to find a focal day of this individual in this time period. However, this would restrain the possible number of cases analysed. An alternative method is to take advantage of the fact that chimpanzees that are part of the same parties, from which a focal individual is chosen, typically engage in the same activity as the focal. In effect, we found that the activity of members of a party was identical to the one of the focal individual in 80% of 31,278 scans. We concluded that this method could both provide a larger dataset to analyse, as well as give an accurate picture of how the test subject had behaved during the preceding weeks of the experiment. We now clarify this approach in the Results (first paragraph) and provide additional details on this analysis in the Methods (subsection “Observational data”, first paragraph).

3) The authors' need to clarify if experimental subjects were tested alone (as suggested in the fifth paragraph of the Introduction), and if so, how this was achieved given the description provided in the last paragraph of the subsection “The ‘honey-trap’ experiment”, which seems to indicate that other group members may have been present.

We apologize for the misunderstanding induced by our phrasing. While our set-up aimed to test experimental subjects when they were alone, we could not prevent them from being joined by other individuals in some cases. This is because the experimenter could not intervene in anyway once a given individual had found the honey-trap apparatus and started engaging with it to ensure that no chimpanzee would be able to connect the honey provided with humans, a point of crucial importance in the context of field experiments. Additionally, to test mothers with dependent offspring, or the offspring themselves, we could only aim for particular family units, as they would always remain together. As a consequence, our dataset consists of trials recorded alone, in familial or social settings. We recorded a total of 124 cases out of 292 (42.5%) where the tested subjects were strictly alone and 86 cases out of 292 (29.5%) where we tested individuals within a family unit (a total of 72%). Finally, in 82 cases out of 292 (28%), other individuals joined the experimental subject in the course of the experiment and also engaged with the honey-trap experiment. We have rewritten the original sentence to avoid confusion (Introduction, eighth paragraph). We have also added information about the social context of the experiment in the Methods and provide a new figure (subsection “The ‘honey-trap’ experiment”, second paragraph and Figure 3). Regarding tool use, 6 instances of tool use occurred in the alone setting, 7 in the family setting, and 8 in the social setting, including 3 when another test subject had previously used a tool. However, because it is unclear whether individuals influenced each other in using tools (discussed in Gruber, 2016), we consider these data points as independent. We have modified the original text (in the aforementioned paragraph).

4) The Sonso chimpanzee community in Budongo have an unusual ecology-showing little evidence of a true low-food availability season (Newton-Fisher 1999). Some key behavioral responses of chimpanzees to low food availability also appear to be absent at Budongo, including the positive relationship between food availability and party size (Newton-Fisher et al. 2000). At many other study sites, chimpanzees respond to decreases in food availability via increased sub-grouping. This raises the question of whether the authors have actually tested patterns of tool-use under true periods of energetic stress. The fact that% time traveling and% fruit in diet were not significantly collinear in this study (subsection “Tool use model”, second paragraph), suggests that they may not have, as chimpanzees tend to range further when fruit availability is low. Given these peculiarities of their study community, the authors need to address more clearly why they believe they tested the 'necessity hypothesis' (i.e. sampled true periods of resource scarcity), as opposed to, sampling one part of what is actually an inverted U-shaped distribution.

We thank the reviewers for this comment that allows us to address variation in feeding behaviour in Budongo Forest across recent years. We are aware of the particular ecology of Budongo Forest, which has for long been considered as preventing food scarcity for its resident chimpanzee population, as shown by Newton-Fisher (1999). Nevertheless, more recent studies have shown that the food offer in Budongo Forest has largely decreased over the last decade, in part due to anthropogenic activities in and adjacent to the forest (Babweteera et al. 2012; Reynolds et al., 2015). As such, Budongo Forest may not have offered the same stable food supply during our study as it used to do in the 1990s. We illustrate this point in the Methods (subsection “Influence of seasonality and identification of periods of food scarcity in Budongo Forest”) and discuss it in the Discussion (second paragraph).

In addition, the focus of our study lies in addressing periods of relative food scarcity specific to the environment the Sonso chimpanzees live in, rather than in ‘absolute’ periods possibly faced by other chimpanzee communities. It is unclear to us how “true” periods of food scarcity can be defined across communities: a savanna chimpanzee faces very different ecological conditions compared to a Sonso chimpanzee but both will certainly face fluctuations in food availability over time. As we now stress in the Introduction, echoing Sanz & Morgan (2013), it is unclear at which point this variation may constitute a real scarcity (Introduction, fifth paragraph). Our analysis is designed to allow testing periods where food supply is relatively less compared to other periods where food is relatively abundant. In fact, taking this temporal variation into account (via our analysis design) is of crucial importance to understand interactions with the apparatus. In this sense, by introducing two new figures, we wish to highlight temporal variation not only in feeding behaviour (Figure 4), but also in the relationship between feeding behaviour and travel time and party size (Figure 5).

In effect, we observe a large variability in terms of ripe fruit feeding and fig feeding (taken as a proxy for fallback food, see reply to comment 5) across the years of the study in the figures now provided in complement to the original Figure 3 (now Figure 4). We also see extensive variation with respect to the mentioned relationship between ripe fruit consumption and party size, with the full extent of possible situations: in some years, party size correlates with ripe fruit feeding (2009, 2013 and 2014), suggesting the possible visible effect of food scarcity usually outlined by other studies (Figure 5). However, although smaller party sizes have been taken as evidence of food scarcity, this correlation is not found in all chimpanzee field sites. For instance, in a recent study at Goualougo, researchers found that party sizes remained relatively stable across the year while food availability itself was not (Sanz & Morgan 2013). We also found this pattern for some years (2008, 2012), or even a negative relationship between party size and ripe fruit feeding (2010, 2011, Figure 5). This suggests that our data encompass a large range of possible cases and therefore, that they are well suited to test the effect of the variation of food offer and travel on tool use. Figure 4 and Figure 5 thus now highlight that the experiment was conducted taking into account this variation. We also wish to point out that absence of problematic collinearity in our model does not indicate that a relationship between any two predictor variables is exactly zero. In fact, in Sonso, we find that the relationship between ripe fruit feeding and travel time varies dramatically within and between years (Figure 5).

*5) How do the authors know that longer travel times are in fact evidence of energy stress? While travel will have some inherent energy cost, chimpanzees may be more active when food is more plentiful, and may be more likely to pursue attractive but less energy-rich foods when food energy is plentiful. For example, it has been suggested that chimps hunt more often when food availability and group sizes are high. Recent work from Tai forest (Riedel 2011) finds that high ranking females are more gregarious, eat a higher quality diet, and travel further than low ranking females. These studies suggest that high investment in travel is not necessarily a marker of energetic need. Similarly, because of the ecology of the Sonso community, it is not necessarily clear that the% of time spent feeding on ripe fruit is necessarily indicative of energetic need. In fact, if fruit is abundant, it may take individuals less time to meet their energetic needs. Demonstrating an increase in the amount of time spent feeding on fallback foods would provide more convincing evidence of energetic stress. Basically, you need to convince the reader that your proxies for energetic need are valid.*

We agree with the reviewers that longer travel times are not necessarily evidence of energy stress. While studies have shown convincingly that terrestrial travel in chimpanzees is costly compared to other activities (e.g. Pontzer et al. 2014), the examples provided by the reviewers show that longer travels are not necessarily markers of energy stress, as long energy balance remains positive. Indeed, if walking costs are compensated by the possibility of fulfilling one’s normal energetic costs or by reaching a high-praised food reward through travel, there is no reason to assume that travel is a necessarily costly activity. The example of high ranking females at Tai illustrates this: the high ranking females are more gregarious, support each other, and can easily monopolize the best food resources, which will fulfil their energetic needs (see Discussion, second paragraph). Similarly, being assured that they will fulfil their energetic needs for the day independently of whether there will be a catch at the end or not, allows chimpanzees to invest energy into hunting, an activity that can be costly both in terms of energy and safety, but that can bring strong social and political effects.

Nevertheless, it is unclear whether these two cases can truly compare with the case described in the current study. Our analyses show that it is a combination of both low food availability and extended travel that lead chimpanzees to engage more with the apparatus, not only one factor. When the food availability was high or travel time short, there was no incentive to engage with the apparatus. In fact, in situations where time devoted to travel was particularly short, the engagement with the apparatus was the lowest (Figure 1). Conversely, when food was readily available and within a short distance, investigating the apparatus while not necessarily being able to reach the honey may reveal itself less profitable than heading straight for the available food (Discussion, first three paragraphs).

Secondly, as already pointed out above in response to point 4), the ecology of Budongo Forest, while possibly less likely to expose its resident primate populations to long periods of food scarcity, has nevertheless changed over the last decades (Babweteera et al. 2012). Relying on an analysis of fallback foods to identify periods of food scarcity in Budongo Forest may however be difficult considering that the diet of the Sonso chimpanzees is mainly constituted of Figs (Newton-Fisher, 1999), which are commonly seen as fallback resources (Marshall & Wrangham 2007, Harrison & Marshall 2011). Nevertheless, we now provide two additional panels (B) and (C) to Figure 4 (formerly Figure 3) to illustrate the variation in ripe fruit feeding and fig feeding in Sonso over the period during which our experiments and observations were carried out. Similar to panel (A), which illustrates seasonality during this period, these two new panels show that our experimental months encompassed a large variation, from months where fruit, respectively fig, consumption was lower than average, to months when fruit, respectively fig, consumption was higher than average. If, figs are considered a fallback food as in other chimpanzee communities, months with higher fig consumption in Sonso can qualify as periods of scarcity. Nevertheless, we would like to stress again that the main goal of our study was to study the influence of a whole range of possible ecological conditions that could influence tool use, and not only periods of possible food scarcity (subsection “Influence of seasonality and identification of periods of food scarcity in Budongo Forest”). As such, we were most interested in variation in food availability. As shown in our updated Figure 4, chimpanzee ripe fruit consumption is characterized by its large variability, ranging monthly from 0.0 to 93.3% of their diet. The effects of this relative abundance and large variability suggest that a combination of both low food availability and high travel to reach it, arguably a good indicator of relative food scarcity in the forest, explains best engagement with the honey-trap experiment.